# Protocadherin 19 Clustering Epilepsy and Neurosteroids: Opportunities for Intervention

**DOI:** 10.3390/ijms22189769

**Published:** 2021-09-09

**Authors:** Rebekah de Nys, Raman Kumar, Jozef Gecz

**Affiliations:** 1Adelaide Medical School, The University of Adelaide, Adelaide, SA 5000, Australia; rebekah.denys@student.adelaide.edu.au (R.d.N.); raman.sharma@adelaide.edu.au (R.K.); 2Robinson Research Institute, The University of Adelaide, Adelaide, SA 5006, Australia; 3School of Biological Sciences, The University of Adelaide, Adelaide, SA 5005, Australia; 4South Australian Health and Medical Research Institute, Adelaide, SA 5000, Australia

**Keywords:** epilepsy, variant, protocadherin, hormones, nuclear hormone receptor, neurosteroids, estrogen receptors, progesterone receptor, androgen receptor

## Abstract

Steroids yield great influence on neurological development through nuclear hormone receptor (NHR)-mediated gene regulation. We recently reported that cell adhesion molecule protocadherin 19 (encoded by the *PCDH19* gene) is involved in the coregulation of steroid receptor activity on gene expression. *PCDH19* variants cause early-onset developmental epileptic encephalopathy clustering epilepsy (CE), with altered steroidogenesis and NHR-related gene expression being identified in these individuals. The implication of hormonal pathways in CE pathogenesis has led to the investigation of various steroid-based antiepileptic drugs in the treatment of this disorder, with mixed results so far. Therefore, there are many unmet challenges in assessing the antiseizure targets and efficiency of steroid-based therapeutics for CE. We review and assess the evidence for and against the implication of neurosteroids in the pathogenesis of CE and in view of their possible clinical benefit.

## 1. Introduction

PCDH19 clustering epilepsy (CE, previously known as girls clustering epilepsy, GCE; female-limited epilepsy, FE; or epilepsy and mental retardation limited to females, EFMR: OMIM #300088) is an X-linked encephalopathy with an incidence rate of 1 per 20,600 live-born females and is characterized by seizures with an average onset of 11.9 months [1,2,3,4,5]. These seizures occur in clusters and generally reduce in frequency by adolescence [1]. CE individuals are often affected by psychiatric comorbidities such as autism spectrum disorder (ASD), hyperactive and/or attention-deficit disorder (ADHD), and behavioural disturbances [1,2]. CE is caused by heterozygous pathogenic variants of the *Protocadherin 19* (*PCDH19*) gene [3]. Despite this disorder being X-linked, CE affects heterozygous females and postzygotic mosaic males, while hemizygous males are asymptomatic carriers [4,5]. Cellular interference remains the best model to explain this unusual pattern of inheritance [3]. This model is supported by mouse and cellular studies that show altered cell sorting in the developing cortex of *Pcdh19^WT/KO^* female mice and increased network activity in mixed cell cultures containing 1:1 *Pcdh19^WT/WT^* and *Pcdh19^KO/KO^* mouse hippocampal neurons when compared to cultures only containing WT or KO cells [6,7]. Furthermore, recent data from *Pcdh19* knockout mice show that *Pcdh19^KO/WT^* female but not *Pcdh19^KO/Y^* mice exhibit mossy fibre presynaptic dysfunction and cognitive impairment [8]. Though *PCDH19^MT/MT^* girls have not yet been identified, the cellular interreference hypothesis contends that these girls may be unaffected, like transmitting males. However, recent studies on *Pcdh19^KO/KO^* female mice showed a lower seizure threshold when compared to that of *Pcdh19^WT/WT^* female mice [9]. These data suggest that cellular interference may not be sufficient to fully explain the mechanisms underlying CE.

The *PCDH19* gene encodes for the PCDH19 protein, which belongs to the 𝛿2 subclass of the nonclustered Pcdh family. PCDH19 is expressed in neurons and glial cells during embryonic development and adulthood and is regulated by miR-99b-3p and T-box brain protein 2 (TBR2) [10,11,12,13]. Like other protocadherins, PCDH19 influences cell adhesion and actin cytoskeleton dynamics [7,10,14,15,16]. PCDH19 also regulates neural morphological maturation and migration through binding the γ-aminobutyric acid type A receptors (GABA_A_Rs) and regulating GABA_A_R subunit surface levels [17]. *PCDH19* downregulation reduces GABA_A_R-mediated tonic currents and neuronal hyperexcitability [18]. Taken together, differences in PCDH19 function may be due to the binding of interacting proteins to the cytoplasmic or extracellular domain of PCDH19 (for example, N-cadherin binds the PCDH19 extracellular cadherin repeats while GABA_A_R alpha 1 binds the cytoplasmic domain) [14,17]. However, recent evidence indicates a new role for PCDH19 in the nucleus, and potentially implicates steroid signalling pathways in CE pathogenesis. This area of research is the focus of this critical review.

## 2. Role of Steroids in CE

### 2.1. Clinical Evidence

Neurosteroids (synthesized in the cortex and hippocampus) and corticosteroids (synthesized in the adrenal glands) [19,20] can influence neurotransmission through binding the GABA_A_R complex and altering chloride influx [21]. Alterations in individual steroid levels are linked to anxiety, stress, depression, and seizure susceptibility [20,22]. In general, neurosteroids such as androgens, and progesterone and its metabolites have anticonvulsant properties (Figure 1a–c), while those such as oestrogen have proconvulsant properties [22] (Figure 1d). Altered basal and postseizure levels of cortisol (Figure 1e, a corticosteroid produced in response to stress) were observed in some adult epileptic patients [23]. The influence of hormones on seizure susceptibility is most obvious in the case of catamenial epilepsy, which refers to changes in seizure frequency relating to alterations in circulating progesterone and oestradiol levels during the menstrual cycle [24]. 

Gene-expression studies of primary skin fibroblasts from patients with CE indicated a possible deregulation of nuclear hormone receptor (NHR)-regulated gene expression in CE girls and likely also boys (more in Section 2.2 below) [25]. Subsequent follow-up studies using blood samples from seizure-active CE girls from multiple cohorts showed lower levels of progesterone metabolite allopregnanolone (which has anticonvulsant properties) than that in normal individuals of the same age [25,26]. Though a link between steroids and seizure patterns has long been known [27], these studies were the first to directly show altered steroid levels in CE.

Steroid levels fluctuate during specific stages of normal human development, leading to sex-specific differences in a range of traits [29]. In epilepsy patients, these fluctuations in steroid levels during puberty and menopause are linked to changes in seizure frequency [30,31]. CE seizure onset and offset generally correlate with changes in sex steroid levels during minipuberty and puberty (Figure 2a) [25]. Sex steroid levels and their neuroprotective effects falling below a threshold after minipuberty may, therefore, open an opportunity for various stressors, such as infection and associated fever, to facilitate the onset of CE. The onset of seizures with low hormonal levels during this developmental period might also be simply coincidental, which is yet to be determined. Seizures often offset when sex steroid levels rise above a threshold during puberty, and thus may provide an anticonvulsant protective effect [25]. To determine if CE girls are deficient in other steroids during development, Trivisano et al. (2017) [26] investigated the steroid levels of pre- and postpubertal CE girls. Although they observed a significant reduction in cortisol and pregnenolone sulfate (PS) levels in postpubertal CE girls (Figure 1e,f), prepubertal CE girls showed no difference in hormonal levels when compared to those of controls. The stimulation of adrenal steroidogenesis through adrenocorticotropic hormone (ACTH) administration to pre- or postpubertal CE girls did not increase PS levels to that of controls. Furthermore, an increase in cortisol levels due to ACTH stimulation was less sustained in prepubertal CE girls than that in the controls [26]. This study shows that there is an abnormal adrenal response in CE individuals throughout development and changes in steroid levels postpuberty. Taken together, these investigations strongly suggest alerted steroidogenesis in seizure-active CE girls. However, steroid-level variations during CE seizure onset and offset have not yet been investigated.

Although seizure onset in CE mosaic males coincides with a decrease in oestradiol levels, seizure offset cannot be correlated with an increase in steroid levels at present due to the small number and young age of male patients (Figure 2b) [32]. As more mosaic males are identified, it is interesting to see if seizure offset follows the same pattern as that in the affected females. One possibility that should be investigated is the effect of androgen levels on CE seizure onset and offset. Androgens have anticonvulsant properties, although evidence is inconsistent. This may be due to the conversion of testosterone into various metabolites that have pro- or anticonvulsant properties [22]. Men with epilepsy often have low levels of serum-free testosterone, and may experience abnormalities relating to reproduction and gonadal function [33]. Steroid levels have not yet been systematically measured in mosaic males, and androgen levels have not been measured in any CE patients, which opens an opportunity for further investigation. Therefore, the effect of steroids on seizure onset and offset in mosaic males cannot yet be confirmed or ruled out.

### 2.2. Gene-Expression Evidence

Steroids can regulate neural gene expression by binding to NHRs [34]. This occurs through either genomic or nongenomic mechanisms [35]. The microarray-based gene expression analysis of CE patient fibroblasts identified significantly dysregulated genes involved in cellular processes such as cell-to-cell signalling, morphology, growth, proliferation, and development [25]. Interestingly, 22% of the dysregulated genes identified were bona fide targets of the NHRs’ progesterone receptor (PGR), chorionic gonadotropin (Cg), and oestrogen receptor (ER) α. Many of the sex-biased dysregulated genes identified in CE girls had expression patterns more like those of males than those of female controls. Some of these genes (*oxytocin receptor (OXTR), glutamate receptor 1 (GRIA1)*, and *aldo-keto reductase family 1 member C3 (AKR1C3)*) were also dysregulated in mosaic males, where expression levels were more like female controls [25]. As sex biases in gene expression can occur during embryonic development [36], it is possible that the molecular changes responsible for altered sex-biased gene expression in CE fibroblasts occur in utero long before the first seizure. However, these observations are yet to be independently validated, and in utero gene-expression studies on *Pcdh19^KO/WT^* mice have not been performed.

#### 2.2.1. Aldo-Keto Reductase Family 1 Member C3 (AKR1C3)

The AKR1C superfamily contains four paralogous genes (all located on chromosome 10 in *Homo sapiens*) that play crucial roles in the production and metabolism of steroids and neurosteroids [37]. For example, *AKR1C2* and *AKR1C4* are genetically mutated in 46 XY individuals with disordered sexual development (DSD) [37]. Consequently, the dysregulation of some of the *AKR1C1-4* identified genes could be related to the pathophysiology present in CE individuals. In that regard, the hydroxysteroid dehydrogenase *AKR1C3* was significantly downregulated in the skin fibroblasts of CE girls when compared to that of controls [25]. AKR1C3 is a moonlighting protein, with enzymatic and activator functions among its many roles. AKR1C3 is involved in the metabolism of neurosteroids such as the conversion of oestrone into 17β-oestradiol, progesterone into 20α-hydroxyprogesterone, androstanedione into dihydrotestosterone (DHT), and androstenedione into testosterone, thus influencing the action of their associated NHRs (Figure 3a) [25,38,39,40]. AKR1C3 also interacts with androgen receptor (AR) in a ligand-dependent manner to coactivate AR-mediated gene expression (Figure 3b). These enzymatic and AR activation functions are mediated by different regions of the AKR1C3 protein, with the full-length protein required for enzymatic function, and ammino acids 171–237 required for AR activation [41]. The downregulation of *AKR1C3* could contribute to altered steroidogenesis in CE girls, or influence gene expression through the regulation of PGR and ERα ligand production. Interestingly, unaffected transmitting males, compared to the male controls, were observed to have slight upregulation of *AKR1C3* [25]. It is unknown if this could result in higher serum steroid levels, thereby providing a seizure-protective effect for transmitting males. The dysregulation of *AKR1C3* could explain at least some of the altered steroidogenesis and gene expression seen in CE individuals.

An important question to consider is the cause of *AKR1C3* dysregulation in CE individuals. *AKR1C3* expression is repressed by androgen-bound AR in the prostate (Figure 3c) [40]. At the same time, AKR1C3 is involved in the production of androgens such as DHT and testosterone (Figure 3a). This means that *AKR1C3* could possibly regulate its own transcription through a feedback mechanism, an observation that is yet to be demonstrated. It is, therefore, unknown if *ARK1C3* dysregulation in CE individuals is caused by altered AR-mediated *AKR1C3* regulation through, for example, an overall disruption of the steroid synthesis pathway or a disruption to AR coregulators. This could lead to a perpetual cycle of altered androgen production and AR-mediated *AKR1C3* dysregulation. On the other hand, disruptions in other underlying gene-regulator mechanisms (such as transcription factors or epigenetic regulatory mechanisms) could also cause *AKR1C3* dysregulation, in turn leading to altered steroidogenesis. Although the promoter region of *AKR1C3* contains putative binding sequences for several transcription factors, their regulatory role is yet to be demonstrated [40]. In the case of epigenetic-mediated regulation, *AKR1C3* is unlikely regulated by cytosine–guanine (CpG) methylation, as the *AKR1C3* promoter sequence lacks CpG islands, although its regulation by other epigenetic factors cannot be excluded [40]. Therefore, whether *AKR1C3* dysregulation is a cause or a consequence of altered steroidogenesis in CE remains unknown.

#### 2.2.2. Oxytocin Receptor (OXTR)

Another interesting CE dysregulated gene is *Oxytocin Receptor* (*OXTR)* (located on chromosome 3), which is highly expressed in the brain and associated with social behaviour [42]. *OXTR* has significantly higher expression in CE female primary skin fibroblasts when compared to age- and sex-matched controls [25]. In addition, the overexpression of the recombinant wildtype PCDH19 protein increases *OXTR* mRNA expression, while the overexpression of the CE variant PCDH19 protein suppresses *OXTR* mRNA expression in MCF-7 cells [43]. Like *AKR1C3* dysregulation, *OXTR* could contribute to certain CE phenotypes such as seizure and ASD [2]. In animal models, *Oxtr^–/–^* mice display autism like behaviour and increased seizure susceptibility [44]. The impact of *OXTR* on development is thought to be due to OXTR influencing the GABA switch, thus affecting neuronal GABA transition from depolarizing to hyperpolarizing during brain development. This occurs by the OXTR-mediated upregulation of chloride cotransporter KCC2 activity by promoting its stabilization at the neuronal surface in a very early and narrow window during postnatal development [45]. In humans, reduced *OXTR* expression is linked to disorders that are known comorbidities of CE, such as schizophrenia and ASD [1,42,46,47]. Treatment of ASD children with oxytocin improves social abilities, and this improvement was more pronounced in children with deficits in OXT signalling [48]. Therefore, *OXTR* could present a plausible target in the treatment of CE comorbidities.

#### 2.2.3. Apolipoprotein D (APOD)

Apolipoprotein D (encoded by the *APOD* gene, located on chromosome 3) is a glycoprotein known to bind lipids and steroids such as cholesterol, pregnenolone, testosterone, and oestradiol [49]. *APOD* expression is regulated by hormones, as its promoter region contains the oestrogen, progesterone, and glucocorticoid response elements [50,51]. Like *OXTR*, the overexpression of the wildtype recombinant PCDH19 protein increases *APOD* mRNA expression, while the overexpression of the CE variant PCDH19 protein suppresses *APOD* mRNA levels in MCF-7 cells [43]. While *APOD* is significantly downregulated in CE skin fibroblasts when compared to control females, the increased expression of *APOD* was found in patients affected by a variety of disorders, including schizophrenia, Alzheimer’s disease, Parkinson’s disease, bipolar disorder, and multiple sclerosis [25,49,52]. Knock-out (KO) studies in model organisms indicated a role of ApoD in neuroprotection and oxidative stress response [53,54]. The loss of *Glial Lazarillo (GLaz,* a homolog of *APOD*) in *Drosophila* increased susceptibility to oxidative and starvation stress, reduced lifespan, reduced body-fat physiology, and reduced locomotor activity when under oxidative stress [54]. Likewise, ApoD-KO mice have reduced locomotor activity and increased vulnerability to oxidative stress [53]. Although the effect of reduced *APOD* expression on seizure susceptibility is yet to be investigated, a study by Najyb et al. (2017) found that *ApoD* overexpression in transgenic mice reduced the number and severity of KA-induced seizures. *ApoD* overexpression was also found to reduce KA-induced apoptosis, attenuate the inflammatory process, and decrease cholesterol levels in the cytosolic fraction of the brain. The authors found that *ApoD* is involved in the regulation of cholesterol uptake by hippocampal neurons, which was increased after KA treatment [55]. Therefore, reduced *APOD* expression in CE girls may be caused by alterations in the steroid pathway/NHR-mediated gene regulation and may contribute to the CE phenotype through its roles in neuroprotection.

#### 2.2.4. PCDH19 as a Coregulator of NHR-Mediated Gene Regulation

How could heterozygous variants in *PCDH19,* a gene encoding a protein involved in calcium-dependent cell adhesion, cause such widespread gene dysregulation? The answer to this question comes from our evolving understanding of the function of PCDH19 in the cell. We showed that PCDH19 localizes to the nuclear fraction of MCF-7 breast-cancer cells and interacts with non-POU-domain-containing octamer-binding protein (NONO)/p54nrb (a regulator of steroid hormone receptors) to coregulate oestrogen receptor (ER) α-mediated transcription. ERα-mediated gene transcription is enhanced in the presence of PCDH19 and NONO, but not PCDH19 alone. Interestingly, CE pathogenic variants were unable to enhance ERα-mediated gene transcription [43]. ERα can influence gene transcription not just through the genomic pathway, but also indirectly through cascade signalling from the cell surface (Figure 4) [35]. It is, therefore, possible that PCDH19 plays a role in ERα-mediated cascade activation that emanates from the cell membrane through interactions with other proteins. Furthermore, whether PCDH19 associates with other NHRs (such as PGR and AR) to regulate gene expression is unknown. Taken together, the consequences of pathogenic *PCDH19* variants on NHR-mediated gene dysregulation could be more pronounced than what we currently understand, and therefore need further investigation.

### 2.3. Cellular Evidence

Despite the large body of the published PCDH19 literature, cellular localisation of this protein appears to be an unsettled issue. PCDH19 is widely regarded as a membrane localised protein, as identified in proliferating human induced pluripotent stem cells (iPSCs) and mature neurons [56]. PCDH19 also plays a role in facilitating cell adhesion and aggregation, supporting the concept that PCDH19 is membrane-bound [7,14]. Furthermore, PCDH19 localises to the synapses and presynaptic puncta of mouse hippocampal neurons [6,12]. However, as mentioned above, Pham et al. (2017) used subcellular fractionation to show that PCDH19, along with NONO and ERα, are present in the nuclear fraction of MCF-7 cells. Immunofluorescence studies in HeLa cells, MDCK cells, and mouse hippocampal neurons showed that the C-terminal region of PCDH19 has a predominantly perinuclear and occasional NONO-paraspeckle localisation [43]. These experiments were performed in MCF-7 cancer cells, which may not be the best model for investigating the consequences of PCDH19 variation that causes heterogenous neurological phenotypes [2]. Other immunofluorescence studies in proliferating iPSCs and polarized cells of the neural rosette lumen show that PCDH19 localises to the mitotic spindle pole of the dividing cells and affects its formation, suggesting that PCDH19 may play a role in regulating asymmetric versus symmetric cell division during neurogenesis [56,57]. Taken together, these data indicate differences in the localisation and possible function of PCDH19 in different cell types and at different cell cycle stages. Therefore, the role of PCDH19/ERα-mediated gene regulation in the brain may also vary depending on developmental stage and cell type.

## 3. Opportunities for Intervention

CE seizures are often resistant to currently available antiepileptic drugs (AEDs), making research into discovering new treatment options vital [58]. Currently, the most effective AEDs used in the treatment of CE, clobazam and bromide, have response rates (defined as a seizure reduction of at least 50% after 3 months) of only 68% and 67%, respectively [59]. Although the use of progesterone and 17β-oestradiol for treating epilepsy was extensively studied [60,61,62], steroids are infrequently used in treating CE, and there are conflicting reports on their effectiveness [59,63] (see Table 1). Higurashi et al. (2015) investigated the effect of corticosteroids on reducing seizure frequency in five CE patients [64]. Higurashi et al. (2015) recruited patients before the age of typical spontaneous seizure remission to prevent the overestimation of drug efficacy [64]. This study showed that all CE patients had a reduction in seizure clusters, findings that are supported by those of Bertani et al. (2015), who also observed improvements in social interactions and speech function with therapy [64,65]. However, follow-up analysis suggests that these steroid-based AEDs may have only short-term effects, with seizures often recurring within a few weeks to months after treatment, especially with fever [59,64]. As these studies were performed on a small number of patients and with considerable variation in the experimental design, the results should be interpreted with caution.

Ganaxolone, a synthetic analogue of allopregnanolone, is the only neurosteroid-like agent undergoing human clinical trials for the treatment of epilepsy [67]. The structure of ganaxolone differs from that of allopregnanolone by the presence of a single 3β-methyl group, which eliminates its backconversion by 3α-hydroxysteroid oxidoreductase (3α-HSOR) isoenzymes to the hormonally active intermediate dihydroprogesterone form. Ganaxolone is metabolised to at least 16 different compounds; the primary product, 16𝛼-hydroxyganaxolone, is inactive in the pentylenetetrazol (PTZ)-induced seizure model [68]. Ganaxolone is a positive allosteric modulator of GABA_A_R [69] that reduced induced seizures in mice and rats [70]. Ganaxalone does not bind steroid receptors, which reduces adverse effects [67]. Since its development, ganaxolone underwent many clinical trials for the treatment of adult and paediatric epilepsy [67,71]. Ganaxolone is currently in a Phase 2 placebo-controlled clinical trial by Marinus Pharmaceuticals (Violet Study) for the treatment of CE [72]. Recent preliminary data released from Marinus showed a median 61.5% reduction in seizure frequency for ganaxolone compared to 24.0% for the placebo (*p* = 0.17) [73]. This is an exciting and very promising result.

Though the development of AEDs for the treatment of CE is vital, CE comorbidities are often just as debilitating. CE has a penetrance of 90%, with the severity of seizure clusters and the presence of comorbidities differing among patients [1]. In fact, several cases of discordant monozygotic CE twins were identified, indicating that variable clinical outcomes are not solely due to the underlying genetic differences [74,75,76]. This diverse range of comorbid phenotypes and seizures pose a major barrier in identifying a therapeutic agent that can be used for treating all CE patients [59]. The differences in clinical symptoms could be due to multiple factors, including X-inactivation skewing or individual steroid levels among different patients [77,78]. Consequently, it would be interesting to investigate differences in gene expression and steroid levels between discordant monozygotic twins, which may explain their variable penetrance.

## 4. CE Mouse Model

Animal models of epilepsy play a major role in AED development [79]. Several mouse models of CE exist and greatly contribute to CE research [12,80,81]. However, they do not closely recapitulate the phenotype of CE patients. For example, *Pcdh19^+/β–Geo^* mice (exon 1–3 deleted and replaced with a *β-Galactosidase-Neomycin* fusion cassette) do not display spontaneous seizures [12], though electrocorticogram (ECoG) analysis shows that these mice have increased spike-wave discharge (SWD) events when compared to control *Pcdh19^+/+^* or *Pcdh19^β–Geo/β–Geo^* mice [7]. Furthermore, whereas variable cortical folding abnormalities are a characteristic of CE [7], *Pcdh19^+/β–Geo^* mice do not show any structural brain abnormalities [12]. Female *Pcdh19^+/β–Geo^* and male *Pcdh19^Y/β–Geo^* mice have autism-like phenotypes when compared to wildtype mice [81]. As CE girls are often affected by ASD [1], this may be a good model for this comorbidity. Another *Pchd19^+/KO^* mouse model (exon 1 deleted and replaced with *LacZ-neo*, which was then removed using *Sox-2::Cre* transgenic mouse) displayed normal social interaction and anxiety-like behaviour, but abnormal mobility under stress and decreased fear response [80]. Whether this model also lacked spontaneous seizures was not reported [80]. Another *Pcdh19^+/KO^* mouse model (generated using the CRISPR/Cas9 system with guide RNAs specific for exon 1) showed presynaptic defects at mossy fibre synapses and impairment in cognitive behaviours associated with mossy fibre function (such as pattern completion ability), but not other cognitive tasks [8]. The lack of an obvious CE phenotype in these mouse models is not unusual, as null variants in human and mouse orthologues often result in different phenotypes [82]. Another point to consider is that of differences in steroid levels between mice and humans [83,84]. This could perhaps be partially caused by phylogenetic divergence and functional differences between human and mouse AKR1C enzymes that are responsible for producing androgens, oestrogens, progesterone, and prostaglandins (PGs) [85]. Therefore, the lack of an obvious CE phenotype and differences in the steroid pathways between mice and humans may impact the usefulness of *Pcdh19* mouse models in AED development.

## 5. Conclusions and Challenges

CE is a debilitating disorder in desperate need of novel therapeutic intervention. Recent research into the role of steroids in CE pathogenesis has resulted in the identification of CE gene dysregulation, altered steroidogenesis, and the implication of PCDH19 in NHR-mediated gene regulation. The identification of reduced allopregnanolone levels in the blood of CE patients has led to ganaxolone trials. However, many aspects relating to the mechanism of CE pathogenesis, particularly that can directly or indirectly facilitate development of treatments, remain unexplored. First, interactions between PCDH19 and other NHRs should be investigated to fully understand the extent of PCDH19 influence on the steroid pathways. The levels of other hormones such as testosterone in the blood of CE females and mosaic males, and the effectiveness of long-term hormone-based AED treatment should also be investigated. Second, though this review focuses on the role of steroids and PCDH19 in the pathogenesis of CE, changes in the steroid pathway are linked to a variety of other encephalopathies and mental illnesses [20,22,86]. For example, studies in *Cdkl5*-KO mouse neurons (a model for the X-linked encephalopathy CDKL5 deficiency disorder; CDD) identified neuron defects that could be restored with pregnenolone or pregnenolone-methyl-ether treatment [87]. This lead to CDD patients being included in ganaxolone clinical trials, with results of the Marigold Study showing a significant reduction in major motor seizure frequency when compared to in the placebo [86]. With further research into the role of steroids and their receptors in CE, there is a promise and optimism of tangible outcomes for CE patients.

## Figures and Tables

**Figure 1 ijms-22-09769-f001:**
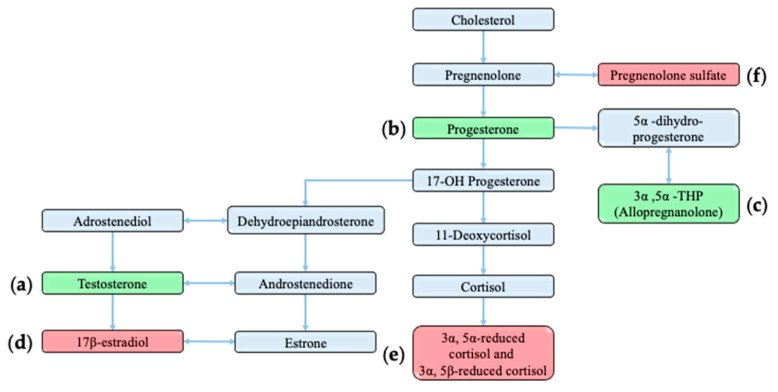
Simplified steroid synthesis pathway. Steroids discussed in this review are highlighted as generally having either excitatory (red) or inhibitory (green) effects on GABA_A_ R. For specific references to (**a**–**f**) panels, please see the main text. Figure modified from [28].

**Figure 2 ijms-22-09769-f002:**
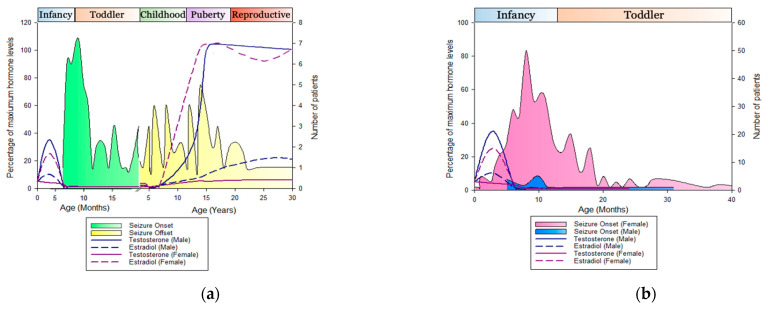
CE seizure onset and offset coincides with fluctuations in sex hormone levels: (**a**) a total of 54 CE females with recorded age of seizure onset and offset (Appendix A) were plotted against average oestradiol and testosterone levels in males and females [29]. CE seizure onset coincided with a decrease in sex hormone levels after minipuberty, while seizure offset coincided with an increase in sex hormone levels during puberty. Figure updated from [25]; (**b**) age of seizure onset for a total of 395 CE females and 22 mosaic males (Appendix A) was plotted against average oestradiol and testosterone levels in males and females during early development. Seizure onset for both CE males and females coincided with a decrease in sex hormone levels.

**Figure 3 ijms-22-09769-f003:**
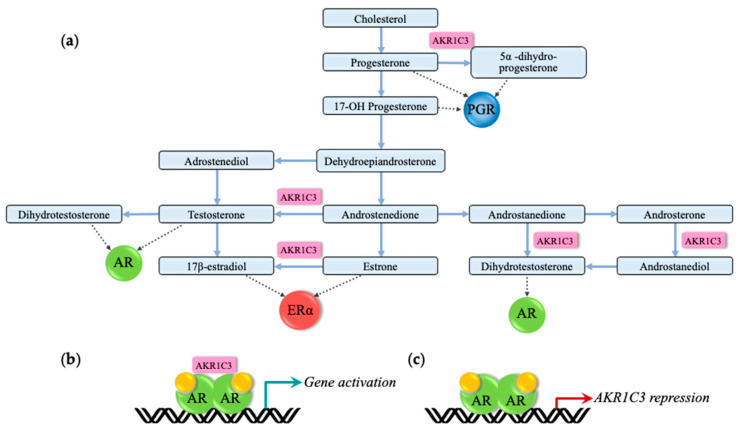
AKR1C3 influence on steroidogenesis and NHR-mediated gene regulation: (**a**) AKR1C3 influences NHR-mediated gene regulation through converting oestrone into the more potent ERα ligand 17β-oestradiol, progesterone into the weaker PGR ligand 20α-dihydroxyprogesterone, androstenedione into the weaker AR ligand testosterone, and androstanedione into the potent AR ligand DHT. Grey arrows represent the NHR corresponding to each ligand; (**b**) *AKR1C3* regulates AR-mediated gene regulation through acting as a coactivator of AR and production of the AR ligands, testosterone, and DHT; (**c**) ligand-bound AR represses *AKR1C3* gene expression [40].

**Figure 4 ijms-22-09769-f004:**
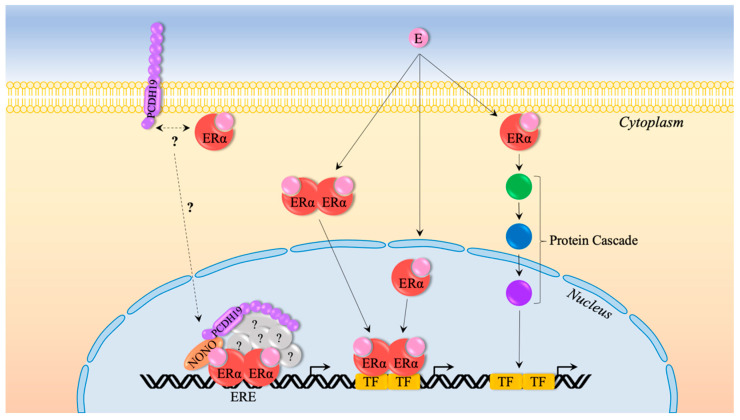
ERα-mediated gene regulation. ERα regulates gene expression through the genomic and nongenomic pathways. Genomic gene regulation involves the binding of oestrogen (E) to cytoplasmic or nuclear localized ERα, resulting in dimerization, a change in receptor confirmation, and, in the case of cytoplasmic ERα, translocation to the nucleus. ERα may recruit corepressors or coactivators (such as PCDH19 and NONO) to regulate gene expression either by directly binding an oestrogen-response element (ERE) or via binding to another transcription factor (TF). The nongenomic pathway involves membrane-bound ERα, rapidly and indirectly influencing gene expression through the cascade activation of DNA binding proteins [35].

**Table 1 ijms-22-09769-t001:** Hormone-based AEDs for treating CE. Investigations into the efficacy of steroids for CE treatment have given mixed results.

Hormone-Based Treatment	No. of Patients	Age	Seizure Reduction (≥50%)	Reference
Corticosteroids	5	10 months–11.6 years	5/5 after the first or second treatment. Clustering recurred within 3 weeks with the onset of fever for 3/5.	[64]
Corticosteroids	1	8 years	1/1.	[64]
Ganaxolone Phase II	11	4–15 years	4/11.	[66]
Steroids	3	Unknown	1/3 after 3 months of treatment. 0/3 after 12 months of treatment.	[59]
Vitamin B6	2	7–8 years	0/2	[63]

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
