# Peer review of "Protocadherin 19 Clustering Epilepsy and Neurosteroids: Opportunities for Intervention"

_ijms, 2021, doi:10.3390/ijms22189769_

Round 1

Reviewer 1 Report

It’s true, that PCDH19-related epilepsy is a rare genetic disease caused by defective function of PCDH19, a calcium-dependent cell-cell adhesion protein of the cadherin superfamily. 

For this reason, review-article is very interesting idea for summary of progress in state of arts in matter such as protocadherin 19 clustering epilepsy and neurosteroids.  

However the following concerns need to be addressed:

In Introduction:

  • All review articles should contain basic methodological information. This review article will be more interesting, if the Authors will describe methods and parameters used in review-paper such as: key words, kind of articles (review or experimental studies, case report, clinical trials), databases, range of years of publication selected to review paper. Moreover, the Authors should inform about progression in number of publications in separate year. These information should be included at the end of introduction or as separate subsection.
  • The authors should write a clear goal for performing a bibliography review. Is this article a critical analysis or a summary of the research results?
  • There is missing information about epidemiology of PCDH19 clustering epilepsy.
  • The Authors should mentioned about subsequent seizure types i.e. grand mal and petit mal epilepsy and point mutations in PCDH19 gene.

In subsection: 2.3. Cellular evidence:

  • According to Cooper et al. [Cooper SR, Jontes JD, Sotomayor M. Structural determinants of adhesion by Protocadherin-19 and implications for its role in epilepsy. Elife. 2016 Oct 26;5:e18529. doi: 10.7554/eLife.18529] over 100 mutations in PCDH19 have been identified in patients with PCDH19-FE, about half of which are missense mutations in the adhesive extracellular domain; neither the mechanism of homophilic adhesion by PCDH19, nor the biochemical effects of missense mutations are understood. In this review-article the Authors should show progress in this opinions.

In this point, the Authors should mentioned results of following papers:

Gursoy S, Ataman E, Baysal BT, Özyılmaz B, Gençpınar P, Hız AS, YiÅŸ U, Ünalp A, Dündar NO, Ülgenalp A, Erçal D. Identification of PCDH19 Gene Mutations/Deletions in Patients with Early Onset Epilepsy. Ann Indian Acad Neurol. 2020 Mar-Apr;23(2):206-210. doi: 10.4103/aian.AIAN_465_19. 

Yang L, Liu J, Su Q, Li Y, Yang X, Xu L, Tong L, Li B. Novel and de novo mutation of PCDH19 in Girls Clustering Epilepsy. Brain Behav. 2019 Dec;9(12):e01455. doi: 10.1002/brb3.1455. 

  • There is not clear for me presence of two subsections, such as 2.2.2. Oxytocin Receptor (OXTR) and 2.2.3. Apolipoprotein D (APOD). The Authors should write about connections between OXTR, APOD and protocadherin 19 clustering epilepsy or neurosteroids (subject of review article).
  • Why didn't the Authors write more about relationship between Protocadherin 19, PCDH19 and neurogenesis?

Lv, X.; Ren, S. Q.; Zhang, X. J.; Shen, Z.; Ghosh, T.; Xianyu, A.; Gao, P.; Li, Z.; Lin, S.; Yu, Y.; Zhang, Q.; Groszer, M.; Shi, S. H., 427 TBR2 coordinates neurogenesis expansion and precise microcircuit organization via Protocadherin 19 in the mammalian cortex. 428 Nat Commun 2019, 10, (1), 3946

Compagnucci, C.; Petrini, S.; Higuraschi, N.; Trivisano, M.; Specchio, N.; Hirose, S.; Bertini, E.; Terracciano, A., Characterizing 527 PCDH19 in human induced pluripotent stem cells (iPSCs) and iPSC-derived developing neurons: emerging role of a protein 528 involved in controlling polarity during neurogenesis. Oncotarget 2015, 6, 26804-26813.

Homan CC, Pederson S, To TH, Tan C, Piltz S, Corbett MA, Wolvetang E, Thomas PQ, Jolly LA, Gecz J. PCDH19 regulation of neural progenitor cell differentiation suggests asynchrony of neurogenesis as a mechanism contributing to PCDH19 Girls Clustering Epilepsy. Neurobiol Dis. 2018 Aug;116:106-119. doi: 10.1016/j.nbd.2018.05.004.

  • At the end of article the authors should include list of abbreviations.
  • There is no citation of new literature and other bibliography which can be a clue for wider discussion in this regards. The following articles should be cited

Borghi R, Magliocca V, Petrini S, Conti LA, Moreno S, Bertini E, Tartaglia M, Compagnucci C. Dissecting the Role of PCDH19 in Clustering Epilepsy by Exploiting Patient-Specific Models of Neurogenesis. J Clin Med. 2021 Jun 23;10(13):2754. doi: 10.3390/jcm10132754.

Samanta D. PCDH19-Related Epilepsy Syndrome: A Comprehensive Clinical Review. Pediatr Neurol. 2020 Apr;105:3-9. doi: 10.1016/j.pediatrneurol.2019.10.009

Gursoy S, Ataman E, Baysal BT, Özyılmaz B, Gençpınar P, Hız AS, YiÅŸ U, Ünalp A, Dündar NO, Ülgenalp A, Erçal D. Identification of PCDH19 Gene Mutations/Deletions in Patients with Early Onset Epilepsy. Ann Indian Acad Neurol. 2020 Mar-Apr;23(2):206-210. doi: 10.4103/aian.AIAN_465_19. 

Yang L, Liu J, Su Q, Li Y, Yang X, Xu L, Tong L, Li B. Novel and de novo mutation of PCDH19 in Girls Clustering Epilepsy. Brain Behav. 2019 Dec;9(12):e01455. doi: 10.1002/brb3.1455. 

Homan CC, Pederson S, To TH, Tan C, Piltz S, Corbett MA, Wolvetang E, Thomas PQ, Jolly LA, Gecz J. PCDH19 regulation of neural progenitor cell differentiation suggests asynchrony of neurogenesis as a mechanism contributing to PCDH19 Girls Clustering Epilepsy. Neurobiol Dis. 2018 Aug;116:106-119. doi: 10.1016/j.nbd.2018.05.004.

Cooper SR, Jontes JD, Sotomayor M. Structural determinants of adhesion by Protocadherin-19 and implications for its role in epilepsy. Elife. 2016 Oct 26;5:e18529. doi: 10.7554/eLife.18529.

Author Response

We would like to thank the reviewers for their critical reading of the manuscript and constructive suggestions.

Reviewer 1:

However, the following concerns need to be addressed:

In Introduction:

  • All review articles should contain basic methodological information. This review article will be more interesting, if the Authors will describe methods and parameters used in review-paper such as: key words, kind of articles (review or experimental studies, case report, clinical trials), databases, range of years of publication selected to review paper. Moreover, the Authors should inform about progression in number of publications in separate year. These information should be included at the end of introduction or as separate subsection.

Authors Reply: We understand that this type of information is required only as part of Systematic Review/s. Our manuscript is not a systematic review of PCDH19-related literature, but instead is focused on reviewing neurosteroids and their role(s) in PCDH19-related clustering epilepsy. However, we have amended the Manuscript as below:

  • Keywords relevant to this study, page 1, lines 24-25;
  • We have now clearly mentioned in the Introduction that the current manuscript is a critical review of neurosteroids in PCDH19-clustering epilepsy and not a systematic review;
  • We reiterate that no specific databases were used for this Review. The published data on the age of the seizure onset and offset, which we used to generate Fig 2, is attached as Supplementary Material, Table S1 and referred to on page 4;
  • All relevant publications used for this review are listed in the References, including some of those suggested below.
  • The authors should write a clear goal for performing a bibliography review. Is this article a critical analysis or a summary of the research results?

Authors Reply: We were the first to implicatePCDH19 gene variation in clustering epilepsy in 2008. The understanding of the role of the PCDH19 protein has so far been limited to and primarily focused on its cell-cell adhesion properties, its role at the synapse and cytoskeletal dynamics. Our Review focuses on a novel and not yet explored aspect of PCDH19 protein function relating to the (dys)regulation of gene expression in association with nuclear hormone receptors. We have clearly stated the aim of this Review in the Introduction; page 2, lines 61-63.

  • There is missing information about epidemiology of PCDH19 clustering epilepsy.

Authors Reply: Very little is known about the incidence of PCDH19-clustering epilepsy. The only study addressing this very issue is that by Symonds et al., 2019 Brain 142: 2303–2318, that reports incidence of epilepsies in Scotland. PCDH19-CE was shown to occur 1 in 20 600 live born females (4.85/100 000;95% CI 1.97-9.15), which is similar to CDKL5 or DEPDC5-related epilepsies. We have included this information and the reference into the revised manuscript.

  • The Authors should mention about subsequent seizure types i.e. grand mal and petit mal epilepsy and point mutations in PCDH19 gene.

Authors Reply: We appreciate the suggestion to go into specific types of seizures seen in PCDH19-clustering epilepsy, but this aspect is beyond the scope of this review and has been reviewed previously by us or our collaborators, see e.g., Kolc et al., 2019 Mol Psychiatry 24:241-251; Trivisano et al., 2018 Epilepsia 59:2260-2271. or Yang et al, 2019 Brain Behav. 9:e01455.

In subsection: 2.3. Cellular evidence:

  • According to Cooper et al. [Cooper SR, Jontes JD, Sotomayor M. Structural determinants of adhesion by Protocadherin-19 and implications for its role in epilepsy. Elife. 2016 Oct 26;5:e18529. doi: 10.7554/eLife.18529] over 100 mutations in PCDH19 have been identified in patients with PCDH19-FE, about half of which are missense mutations in the adhesive extracellular domain; neither the mechanism of homophilic adhesion by PCDH19, nor the biochemical effects of missense mutations are understood. In this review-article the Authors should show progress in this opinions.

In this point, the Authors should mentioned results of following papers:

Gursoy S, Ataman E, Baysal BT, Özyılmaz B, Gençpınar P, Hız AS, YiÅŸ U, Ünalp A, Dündar NO, Ülgenalp A, Erçal D. Identification of PCDH19 Gene Mutations/Deletions in Patients with Early Onset Epilepsy. Ann Indian Acad Neurol. 2020 Mar-Apr;23(2):206-210. doi: 10.4103/aian.AIAN_465_19. 

Yang L, Liu J, Su Q, Li Y, Yang X, Xu L, Tong L, Li B. Novel and de novo mutation of PCDH19 in Girls Clustering Epilepsy. Brain Behav. 2019 Dec;9(12):e01455. doi: 10.1002/brb3.1455. 

Authors Reply: We have recently reviewed our current understanding of cellular aspects of the PCDH19-clustering epilepsy (see Gecz and Thomas, 2020 Curr Opin Genet Dev 65:169-175). We discuss recent findings of Hoshina et al 2021Science372:eaaz3893.) in the mouse model section of this review.

  • There is not clear for me presence of two subsections, such as 2.2.2. Oxytocin Receptor (OXTR) and 2.2.3. Apolipoprotein D (APOD). The Authors should write about connections between OXTR, APOD and protocadherin 19 clustering epilepsy or neurosteroids (subject of review article).

Authors Reply: Our original studies using PCDH19-CE patient skin fibroblasts (Tan et al., 2015; Hum Mol Genet. 24:5250-9) showed oxytocin receptor (OXTR) and apolipoprotein D (APOD) among the most significantly dysregulated genes in the PCDH19-CE individuals. Subsequently, our follow-up studies confirmed OXTR and APOD as downstream dysregulated genes in in vitro assays of PCDH19 wildtype and variant recombinant proteins (Pham et al., 2017 Hum Mol Genet. 26:2042-2052). We discussed APOD as this is the only brain transporter of cholesterol, which is the precursor of neurosteroids in the brain. We discuss OXTR as it is directly regulated by steroid hormones and is implicated in autism, one of the most significant, overlapping comorbidities of PCDH19-CE. We have amended the text to clearly explain the association between OXTR, APOD and neurosteroids while referring to these publications.

  • Why didn't the Authors write more about relationship between Protocadherin 19, PCDH19 and neurogenesis?

Lv, X.; Ren, S. Q.; Zhang, X. J.; Shen, Z.; Ghosh, T.; Xianyu, A.; Gao, P.; Li, Z.; Lin, S.; Yu, Y.; Zhang, Q.; Groszer, M.; Shi, S. H., 427 TBR2 coordinates neurogenesis expansion and precise microcircuit organization via Protocadherin 19 in the mammalian cortex. 428 Nat Commun 2019, 10, (1), 3946

Compagnucci, C.; Petrini, S.; Higuraschi, N.; Trivisano, M.; Specchio, N.; Hirose, S.; Bertini, E.; Terracciano, A., Characterizing 527 PCDH19 in human induced pluripotent stem cells (iPSCs) and iPSC-derived developing neurons: emerging role of a protein 528 involved in controlling polarity during neurogenesis. Oncotarget 2015, 6, 26804-26813.

Homan CC, Pederson S, To TH, Tan C, Piltz S, Corbett MA, Wolvetang E, Thomas PQ, Jolly LA, Gecz J. PCDH19 regulation of neural progenitor cell differentiation suggests asynchrony of neurogenesis as a mechanism contributing to PCDH19 Girls Clustering Epilepsy. Neurobiol Dis. 2018 Aug;116:106-119. doi: 10.1016/j.nbd.2018.05.004.

Authors Reply: For completeness we have added the information relating to the role of PCDH19 in regulating the cell division during neurogenesis.

  • At the end of article the authors should include list of abbreviations.

Authors Reply: We have included list of abbreviations at the end of the main text of the manuscript.

  • There is no citation of new literature and other bibliography which can be a clue for wider discussion in this regards. The following articles should be cited

Borghi R, Magliocca V, Petrini S, Conti LA, Moreno S, Bertini E, Tartaglia M, Compagnucci C. Dissecting the Role of PCDH19 in Clustering Epilepsy by Exploiting Patient-Specific Models of Neurogenesis. J Clin Med. 2021 Jun 23;10(13):2754. doi: 10.3390/jcm10132754.

Samanta D. PCDH19-Related Epilepsy Syndrome: A Comprehensive Clinical Review. Pediatr Neurol. 2020 Apr;105:3-9. doi: 10.1016/j.pediatrneurol.2019.10.009

Gursoy S, Ataman E, Baysal BT, Özyılmaz B, Gençpınar P, Hız AS, YiÅŸ U, Ünalp A, Dündar NO, Ülgenalp A, Erçal D. Identification of PCDH19 Gene Mutations/Deletions in Patients with Early Onset Epilepsy. Ann Indian Acad Neurol. 2020 Mar-Apr;23(2):206-210. doi: 10.4103/aian.AIAN_465_19. 

Yang L, Liu J, Su Q, Li Y, Yang X, Xu L, Tong L, Li B. Novel and de novo mutation of PCDH19 in Girls Clustering Epilepsy. Brain Behav. 2019 Dec;9(12):e01455. doi: 10.1002/brb3.1455. 

Homan CC, Pederson S, To TH, Tan C, Piltz S, Corbett MA, Wolvetang E, Thomas PQ, Jolly LA, Gecz J. PCDH19 regulation of neural progenitor cell differentiation suggests asynchrony of neurogenesis as a mechanism contributing to PCDH19 Girls Clustering Epilepsy. Neurobiol Dis. 2018 Aug;116:106-119. doi: 10.1016/j.nbd.2018.05.004.

Cooper SR, Jontes JD, Sotomayor M. Structural determinants of adhesion by Protocadherin-19 and implications for its role in epilepsy. Elife. 2016 Oct 26;5:e18529. doi: 10.7554/eLife.18529.

Authors Reply: Thank you for these suggestions of works which are from our team or we are well aware of. We have added the following articles to the review, which we consider to be relevant:

Borghi R, Magliocca V, Petrini S, Conti LA, Moreno S, Bertini E, Tartaglia M, Compagnucci C. Dissecting the Role of PCDH19 in Clustering Epilepsy by Exploiting Patient-Specific Models of Neurogenesis. J Clin Med. 2021 Jun 23;10(13):2754. doi: 10.3390/jcm10132754.

Samanta D. PCDH19-Related Epilepsy Syndrome: A Comprehensive Clinical Review. Pediatr Neurol. 2020 Apr;105:3-9. doi: 10.1016/j.pediatrneurol.2019.10.009

Cooper SR, Jontes JD, Sotomayor M. Structural determinants of adhesion by Protocadherin-19 and implications for its role in epilepsy. Elife. 2016 Oct 26;5:e18529. doi: 10.7554/eLife.18529.

Reviewer 2 Report

de Nys et al summarize in this review, the current knowledge on the role of hormone pathways in clustering epilepsy, such as PCDH19. The authors described the implication of neurosteroids in the pathogenesis of CE and their effectiveness as treatment for CE, highlighting also that there is a need to better understand the role of neurosteriods in the pathophysiology of PCDH19.  The paper is well written and easy to understand. The review gives an interesting widespread and scientific perspective on possible opportunities for intervention on PCDH19 patients, therefore, deserves publication in International Journal of Molecular Sciences.

Author Response

We would like to thank the reviewers for their critical reading of the manuscript and constructive suggestions.

Reviewer 2: This reviewer had no specific comments or suggestions. Thank you.